# Conceptual Design, Development, and Preliminary Safety Evaluation of a PWR Dry Storage Module for Spent Nuclear Fuel

**Taehyeon Kim** **, Kiyoung Kim, Donghee Lee, Taehyung Na, Sunghwan Chung and Yongdeog Kim ***

Korea Hydro & Nuclear Power, 70 Yuseongdaero 1312, Daejeon 34101, Korea; taehyeon.kim@khnp.co.kr (T.K.); kiyoungkim@khnp.co.kr (K.K.); donghee.lee@khnp.co.kr (D.L.); taehyung.na@khnp.co.kr (T.N.); eituzed@gmail.com (S.C.)
* Correspondence: yongdkim@khnp.co.kr

**Abstract:** Dry storage systems are one of the storage methods for spent nuclear fuel used in many countries that operate nuclear power plants. To ensure the safe storage of spent nuclear fuel, dry storage systems are designed to maintain radiation shielding, thermal management, and subcritical and mechanical integrity. In addition, these systems must be able to withstand earthquakes, tornadoes, floods, extreme temperatures, and other operating and design-based accident conditions. In order to develop a model with safety and economic feasibility by analyzing various dry storage systems, a vertically dry storage module was proposed, and evaluations were performed on safety evaluation along with design requirements. As a result of the evaluation, all of the safety design requirements were met. This evaluation's results can be used as basic data for the detailed design of the dry storage module to proceed with further research, including the preparation of a safety analysis report and experimental verification for licensing applications.

**Keywords:** spent nuclear fuel; dry storage; pre-safety evaluation; conceptual design; storage module



## 1. Introduction

In countries around the world, the amount of spent nuclear fuel (SNF) for dry storage within nuclear power plant (NPP) sites is increasing significantly, because the operational period of NPPs is getting longer, the number of decommissioned NPPs is increasing, and national SNF management policies are delayed. [1] Due to the increase in the required amount of SNF dry storage systems (SDSSs), there is a tendency to use more efficient and cost-effective systems. Similar to other countries, there is uncertainty about the operation of SNF management facilities in Korea. Additionally, as the loss of full core reserves will come soon, dry storage of SNF at NPP sites should be promoted in a timely manner in with consideration of the stable operation of decommission of NPPs and the optimal SDSS. Therefore, in this study, the design of an SDSS was conceived to secure the following safety functions:

- Prevention of radioactive material leakage;
- Decay heat removal from SF using natural airflow;
- Maintaining subcriticality of SF;
- Minimization of unnecessary exposure of workers and civilians to radiation;
- Minimization of the operating area of dry storage facilities.

In the conceptual design of the dry storage module (DSM), the concept of concentrating and storing the SF inside the rectangular concrete structure was used to secure the safety function. The DSM uses a canister–cylinder dual structure to prevent radioactive material leakage, and it is possible to secure passive heat removal performance and minimize the radiation exposure and operating area by sharing the flow area inside the concrete structure. A new DSM for PWR SNF is being developed akin to the M/K-400 [2] dry

storage module for PHWR SNF at the WOLSONG NPP site. This design concept has been patented in Korea [3].

SDSSs must be evaluated in accordance with the applicable requirements of related domestic and international regulations to ensure the safety of the storage system and retrievability of the SNF. The system must provide subcriticality, adequate heat removal capacity with passive cooling systems, suitable shielding for radiation protection, and have sufficient structural integrity under the design-based conditions of normal operations, off-normal operations, and accidents. In the conceptual design of a new storage cask called KORAD-21, a preliminary evaluation was carried out considering these conditions [4]. The goal of this study is to evaluate the feasibility of conducting detailed design through preliminary evaluation of the critical, shielding, thermal, and structural fields for the conceptual design of a DSM. In order to design a PWR SDSS, preliminary evaluation was carried out in accordance with the requirements of the related regulations of the Korea Nuclear Safety Act [5] and US 10CFR72 [6].

## 2. Design Basis

### 2.1. Design Criteria

The dry storage system for SNF must be designed to maintain the safe storage conditions of SNF, prevent damage to SNF, assure the retrievability of SNF, and withstand the impacts of design-based conditions such as normal, off-normal, and accident conditions. The main design requirements for the dry storage of SNF are in accordance with the relevant regulations of the Korea Nuclear Safety Act, IAEA SSS No.SSG-15 [7], and US 10CFR72. These regulations are declarative, and straightforward technical design requirements are described in USNRC NUREG-2215 [8]/1536, [9]/1567 [10], and ASME Sec. III [11]. The most important design criteria for SDSSs are to ensure that public health and safety are protected from the SDSS, and to maintain the integrity of the SNF throughout its design life under normal storage conditions. Therefore, it is necessary to develop a design that meets the regulatory requirements, establish the design requirements necessary to secure the safety of the SDSS's components, and then check whether they are satisfy the regulations. According to 10CFR72, an SDSS must be able to safely store SNF under normal, off-normal, and accident conditions throughout the storage period, and the retrievability of the SNF must be secured. Normal conditions include SDSS handling, fuel loading and unloading, and ambient temperature changes occurring during normal operation. Off-normal conditions are events that can occur frequently in situations beyond normal operating conditions, such as damage to the confinement boundary, blockage of some cooling channels, human error, and equipment-handling error. Accident conditions are accidents beyond the off-normal conditions, and are events during operations such as falling, overturning, fire, loss of confinement boundary, and complete blockage of the cooling channel, as well as events caused by natural phenomena such as flooding, typhoons, tornadoes, earthquakes, and tsunamis.

The design requirements of the SDSS are presented by the dry storage facility operator, reflecting the operating conditions and regulatory requirements of the facility. The design requirements of the spent fuel dry storage module proposed in this study are to ensure subcritical maintenance, radiation shielding, heat, and structural safety for a design life of 50 years under normal, off-normal, and accident conditions. The detailed design requirements are presented in Table 1.

**Table 1.** Design requirements for the DSM.

| Parameter | | Value |
|---|---|---|
| Fuel Specification | - Max. SNF burnup; <br> - **Initial** enrichment; <br> - Minimum cooling time; <br> - Number of FAs in canister. | 45 GWd/MTU <br> 5 wt.% <br> 10 yrs <br> 24 |
| Subcriticality | - Effective k-value | 0.95 |
| Thermal | Fuel cladding temperature: <br> - Normal; <br> - Off-normal. <br> Concrete temperature: <br> - Normal bulk (local); <br> - Off-normal | 400 °C <br> 570 °C <br><br> 67 °C (93 °C) <br> 176 °C |
| Radiation Shielding | - Surface at module <br> - Surface at pence | 0.5 mSv/h <br> 0.01 mSv/h |
| Seismic | - SSE | 0.3 g |
| Mass | - Total handling mass | 113 t |

FA: fuel assembly; SSE: safe shutdown earthquake.

### 2.2. DSM

Figure 1 shows a schematic diagram of the DSM. Our DSM is a vault-type vertical storage module made of reinforced concrete. The horizontal concrete modules or vertical concrete casks currently in commercial use occupy a larger storage area than the vertical concrete storage module. Moreover, the vertical concrete storage module has the advantage of reducing the storage area by about 30% due to these structural characteristics. It is advantageous for maintenance due to applying a replaceable design to the cylinders installed in the module. In addition, this DSM provides improved seismic safety due to the wide lower cross-sectional area in contact with the ground. The air inlet and outlet of the SDSS are located at the top and bottom of the side wall, respectively, and meet the requirements of the dry storage system through non-powered natural convection cooling with external air. The positions of the air inlet and outlet are arranged so that they intersect with respect to the cylinder's position. This arrangement is advantageous for cooling the canister, because the air introduced at the inlet makes a flow that is discharged after contacting the cylinder as much as possible. The upper part of the DSM has no air inlet/outlet so as to secure operational convenience, and the short side minimizes interference due to multiple arrangement. The dimensions of the DSM were determined by the results of critical, shielding, thermal, and structural analysis, with width and height of $20.5 \times 10 \times 7.8$ m, respectively.

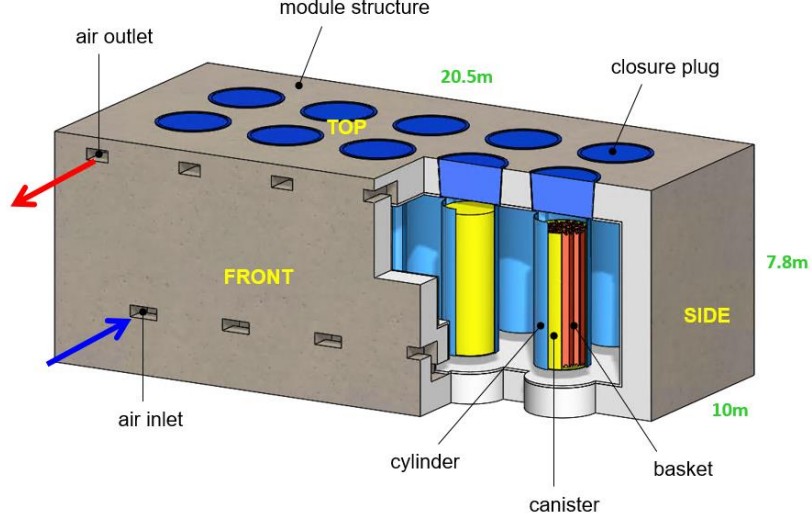

**Figure 1.** Schematic diagram of the DSM.

### 3. Safety Evaluation of the DSM

According to NUREG-2215, the certificate of compliance (CoC) of the dry storage system must submit the evaluation results for structure, heat, shielding, criticality, materials, containment, and accident conditions, along with design requirements. The safety analysis report (SAR) for dry storage casks with a recent general license includes these evaluation results. In this study, for the conceptual design of a DSM based on an MPC-24 canister, the fuel type and source term were determined, and critical evaluation for normal conditions was performed. In addition, the surface dose rate requirement of the DSM was determined to be 0.5 mSv/h, and the thickness of the concrete structure and the shape of the entrance/exit area were determined through shielding evaluation. For the shape of the DSM derived from this, thermal evaluation and structural evaluation reflecting normal, off-normal, and accident conditions were performed.

### 3.1. Fuel Type and Source Term

Korea has various types of nuclear power plants, and the types of fuel used are also diverse. Thus, it is necessary to select a representative fuel, and the most widely used PLUS7 fuel assembly in Korea was selected for the calculation of the source term. The source terms to be considered in this safety evaluation were gamma rays and neutron rays, and were calculated using the ORIGEN-S module of SCALE 6.1 [12]. Among the source terms, the gamma rays caused by the nuclear fuel assembly's structural material radiation were also taken into account for the evaluation and calculated using the relative flux fraction of each structural material's area to the neutron flux in the effective fuel area. The characteristic SNF value used in the source term evaluation was 5 wt.% of initial uranium-235 enrichment; the burnup was 45GWd/MTU, and the cooling period was evaluated for 5 years, 10 years and 15 years. Source term values and decay heat output values of the effective fuel area for each cooling period were used as input data for shielding and thermal analyses. The axial profile was not considered in the decay heat calculation, and the decay heat was calculated by summing the calorific values of the light element, actinide, and fission products. Figure 2 shows the decay heat and radioactivity according to the cooling period.

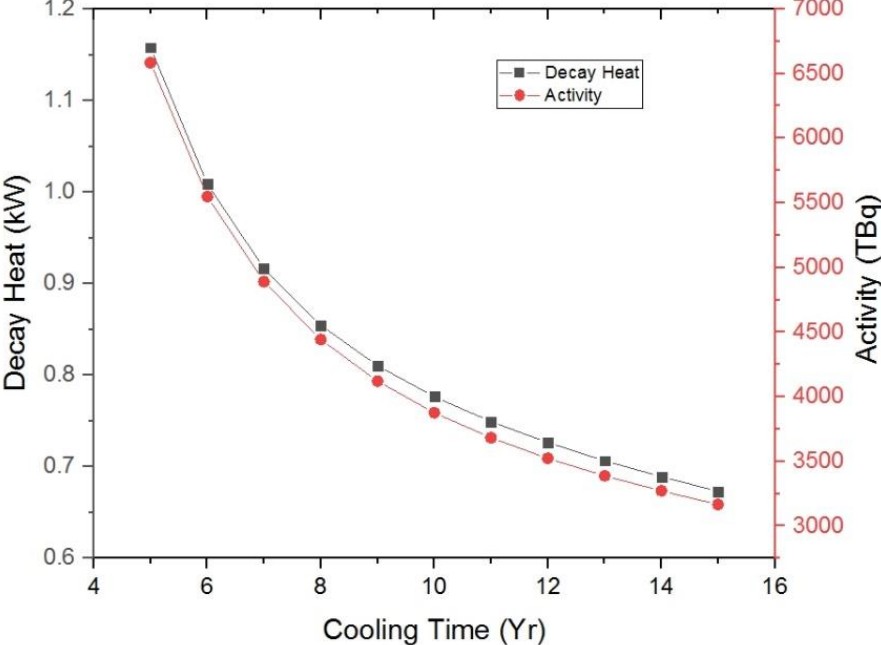

**Figure 2.** Decay heat and activity of the PLUS7 fuel assembly.

### 3.2. Criticality Analysis

The design of fuel canisters in the DSM ensures that the fuel will remain in a configuration that has been predetermined to be subcritical during loading, storage, and retrieval. This criticality design should allow for any consequences likely to result from redistribution or the intrusion of a moderator as a consequence of an internal or external event. The technical standards for criticality evaluation are in accordance with the US 10CFR72.124 "Criteria for Nuclear Criticality Safety", the American Nuclear Society regulatory guidelines ANSI/ANS-57.9 [13], and the US NRC standard review guidelines NUREG-1536. The technical standards for critical evaluation applied to the conceptual SDSS design are as follows:

- $K_{eff}$ (effective neutron multiplication factor) at the 95% probability and 95% confidence level should not exceed 0.95 under the normal, off-normal, and accident conditions.
- The criticality safety of the DSM design should be ensured on the basis of favorable geometry, with a fixed neutron absorber. Where solid neutron absorbers are used, the canister design should provide for a positive means to verify their continued efficacy during the storage. The neutron absorbers' continued efficacy may be confirmed by verification before use, showing that the degradation of these materials cannot occur over the lifetime of the facility.
- The criticality safety of the canister design should not rely on credit for more than 75% of the neutron poison material in the fixed neutron absorber. Comprehensive tests are needed in order to verify the presence and uniformity of the neutron absorber.

In short, critical evaluation should be designed to maintain subcriticality in consideration of normal, off-normal, and accident conditions according to the technical standards presented above, but in this analysis it was designed in consideration of only the full flood condition, as the most conservative assumption was considered. The canister model used for critical evaluation was a 24 FA container, which consists of a basket cell, a neutron absorber, and a neutron-absorber cover. For critical evaluation, the SCALE6.1.3 program developed by ORNL was used, and the library of the ENDF 238 group was applied. Validation of the SCALE6.1.3 program was carried out in accordance with the NUREG-6698 methodology based on the 236 Benchmark Experiments presented by the OECD/NEA. These uncertainties may be statistically combined, and the calculation for the maximum $K_{eff}$ may therefore be described as follows:

$$Maximum\ K_{eff} = K_{cal} + K_{bias} + K_{uncertainty}$$

where: $K_{cal}$ is the calculated SCALE6.1.3 reactivity in three dimensions;

$K_{bias}$ includes the bias from the 236 benchmark calculations;

$K_{uncertainty}$ includes the effects of all uncertainties combined statistically.

$K_{uncertainty}$ includes uncertainties of the manufacturing tolerances of the fuel assembly—such as UO2 density and cladding thickness—and the concrete module, such as canister thickness and the B10 density of the neutron absorber. As a result of the validation of the SCALE codes, bias and uncertainty were assessed as 0.00749 and 0.01079, respectively. Figure 3 shows the radial and axial models used in the evaluation, while Table 2 shows the results of the critical assessment, indicating that the maximum $K_{eff}$ for normal conditions at the 95% probability and 95% confidence level is 0.94153, which is below the design basis of 0.95.

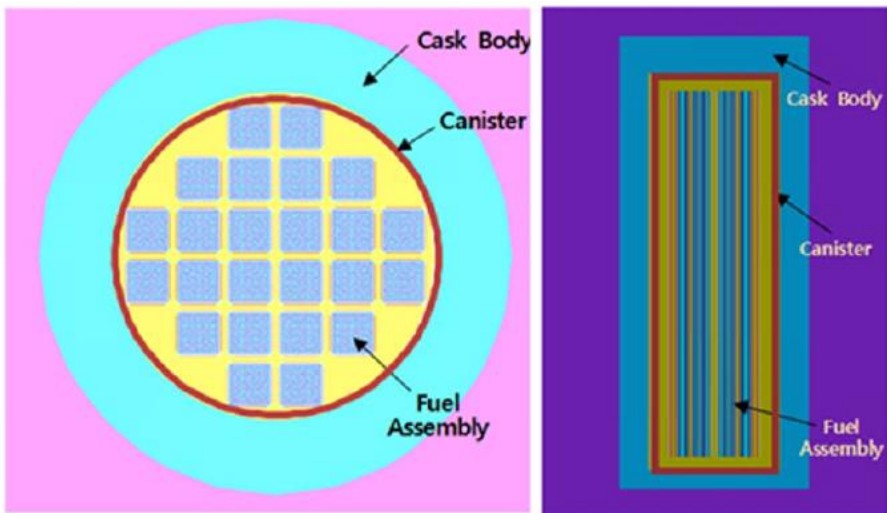

**Figure 3.** Criticality analysis model of the DSM.

**Table 2.** Maximum $K_{eff}$ of the DSM.

| Item | $K_{eff}$ |
|:---:|:---:|
| $K_{cal}$ | 0.92325 |
| Bias | 0.00749 |
| Uncertainty | 0.01079 |
| Maximum $K_{eff}$ | 0.94153 |

*3.3. Radiation Shielding Analysis*

Various structures and concrete walls of the DSM not only maintain its structural integrity, but also function to reduce the exposure of workers to radiation during loading and unloading of SNF, according to the ALARA principle. Radiation shielding analysis determined that the concrete thickness meets the allowable surface dose rate standard of the DSM outer wall, and then calculated the radiation dose rate at the air inlet/outlet. In particular, the inlet/outlet radiation dose rate analysis derived the shape of the air inlet/outlet flow path in the form of steps that can reduce the exposure of workers in dry storage facilities. MCNP 6.0 [14], based on the Monte Carlo method, was used for the radiation shielding analysis, and the assumptions used in this analysis were as follows:

- A homogenization model was used for the effective nuclear fuel area and upper/lower structures of the SNF;
- The axial burn-up profile was not considered;
- A sufficient air layer for the calculation of scattered radiation due to collision between radiation and air was considered;
- The flux-to-dose conversion factor ICRP-74 was used [15];
- The SDSS outer wall's allowable surface dose rate was limited to 0.5 mSv/h.

The MCNP model used in the radiation shielding analysis considered the canister (applies to MPC-24), carbon steel cylinder, airflow region, and concrete structure. Figure 4 shows the MCNP model used to calculate the wall thickness of DSM. Figures 5 and 6 show models for calculating the surface radiation dose rate at the air inlet/outlet flow path, respectively. Figure 8 shows the results of the radiation shielding analysis. Since the effect of neutron rays is insignificant, only gamma rays were considered. As a result, the minimum thickness of concrete that satisfied the allowable surface dose rate of 0.5 mSv/h was 60 cm, and the permissible surface radiation dose rate of the inlet and outlet areas was also 0.5 mSv/h. The results of the radiation shielding analysis are summarized in Figure 7 and Table 3.

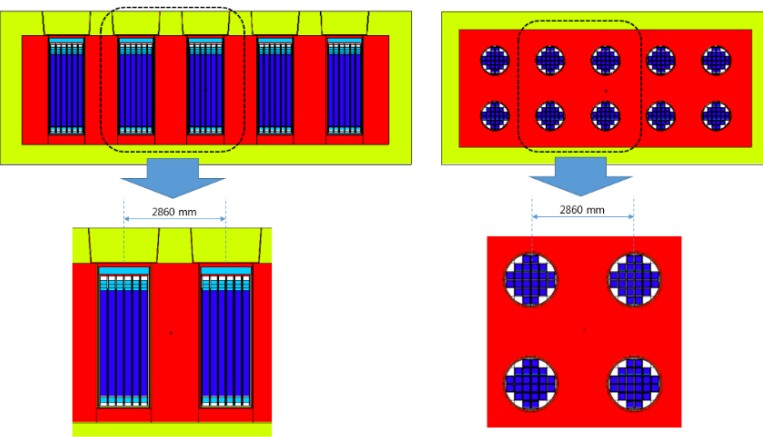

**Figure 4.** Radiation shielding model of the DSM.

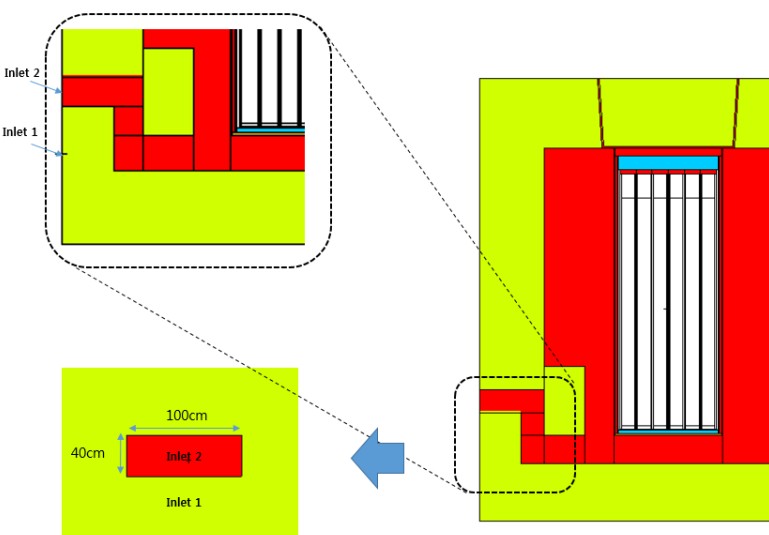

**Figure 5.** Inlet radiation shielding model of the DSM.

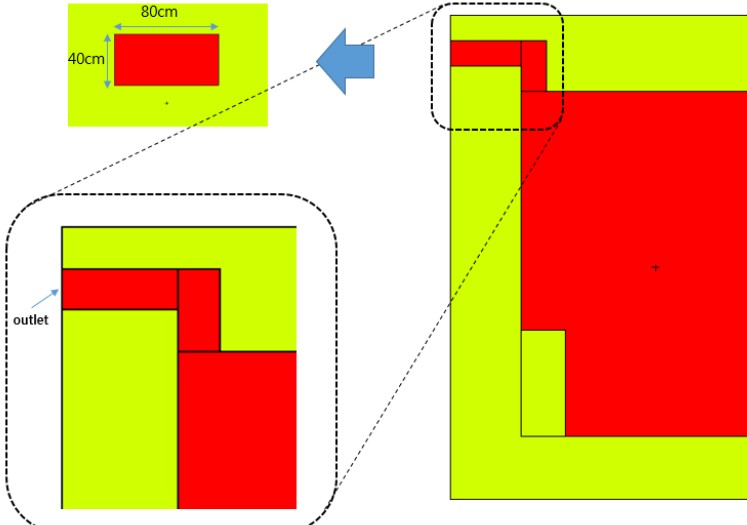

**Figure 6.** Outlet radiation shielding model of the DSM.

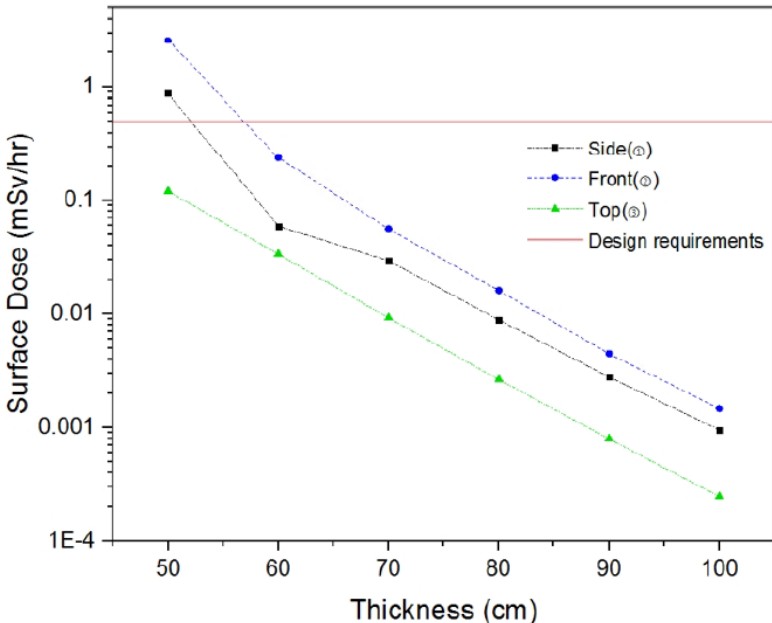

**Figure 7.** Dose rate as a function of concrete thickness.

**Table 3.** Dose rate as a function of inlet/outlet geometry.

| Location | | Neutron (mSv/h) | Gamma Ray (mSv/h) | Total Dose Rate (mSv/h) |
|---|---|---|---|---|
| Inlet (lower) | 1 | $3.293 \times 10^{-6}$ (0.36) | $1.297 \times 10^{-6}$ (0.36) | $4.590 \times 10^{-6}$ |
| | 2 | $4.851 \times 10^{-2}$ (0.42) | $1.315 \times 10^{-2}$ (0.22) | $6.166 \times 10^{-2}$ |
| Outlet (upper) | | $4.92 \times 10^{-2}$ (0.30) | $1.275 \times 10^{-3}$ (0.46) | $5.048 \times 10^{-2}$ |

### 3.4. Thermal Analysis

Thermal evaluation of the DSM was performed to confirm that the temperature of the DSM components and the SNF would be maintained within the limits under normal, off-normal, and accident conditions. It should be ensured that the temperature of the fuel cladding is maintained below the design limit during the storage period to prevent damage to the cladding. According to the requirements of NUREG-1567, with reference to ACI-349 [16], the local maximum temperature of cask concrete structures under normal and off-normal conditions should not exceed 93 °C, and is required to remain below 176 °C under accident conditions. NUREG-1536 requires the maximum temperature of the storage system's containment boundary components to be kept below the storage function maintenance temperature, and requires the fuel cladding pipe's maximum temperature to be kept below 400 °C under normal and off-normal conditions, and 570 °C under accident conditions. NUREG-1536 presents the temperature requirements of the storage system as follows:

- The maximum temperature of the storage boundary components must be maintained below the storage function maintenance temperature.
- Under all conditions, the maximum internal pressure must be maintained below the design pressure.

At this time, the outdoor temperature of the dry storage system is the maximum annual average value of the operating area. In the thermal evaluation of the DSM, the analysis case considered normal conditions at an outside temperature of 40 °C, and off-normal and accident conditions with a 50% and 100% blocked air inlet, respectively.

For thermal evaluation of the DSM, Fluent 19.2 [17]—a commercial computational fluid dynamics (CFD) program—was used. In addition, a conjugation heat-transfer model was used to evaluate the temperature of the airflow region, the concrete wall, and the canister. The heat transfer inside the DSM was evaluated using CFD tools, considering all of the conduction, convection, and radiation heat-transfer mechanisms. To study the effects of mesh resolution and near-wall treatment, three different computational meshes were constructed and evaluated, and a medium-sized mesh was selected. Foam glass insulation (thickness 50 mm) was used between the air layer and the concrete wall to prevent overheating of the concrete structure. The RNG k-epsilon turbulence model was used for the airflow region, and the density of air was set to change according to the temperature, to simulate natural convection. The interpolation scheme for pressure and velocity used a SIMPLE algorithm and upwind scheme. In this CFD calculation, the RMS residual levels for mass and momentum were considered to converge to $1 \times 10^{-4}$, while that for energy was considered to converge to $1 \times 10^{-6}$ or less. The inside of the canister was used to calculate the effective thermal conductivity. Figure 8 shows the calculation model used in the CFD analysis. For efficient CFD analysis, the flow analysis domain used a 1/4 model for normal conditions and a 1/2 model for off-normal and accident conditions. Figures 9–11 show the temperature contours calculated via CFD. The maximum local temperature of the upper inner wall of the DSM was determined to be 90.3 °C in normal conditions, 92.9 °C in off-normal conditions, and 107.4 °C in accident conditions. Therefore, the results of the analysis showed that the integrity of the concrete structure was maintained by satisfying the thermal design criteria under all analysis conditions. However, since the local temperature of 90.3 °C in concrete has a very small margin compared to the design standard under normal conditions, it is necessary to improve the structure by granting it more efficient heat-removal performance through detailed design.

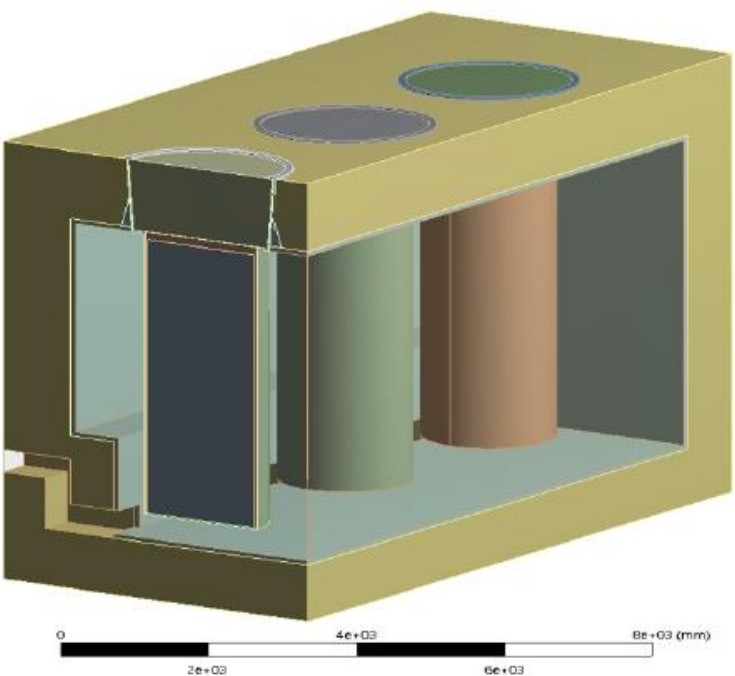

**Figure 8.** Fluid calculation domain for the DSM (quarter model).

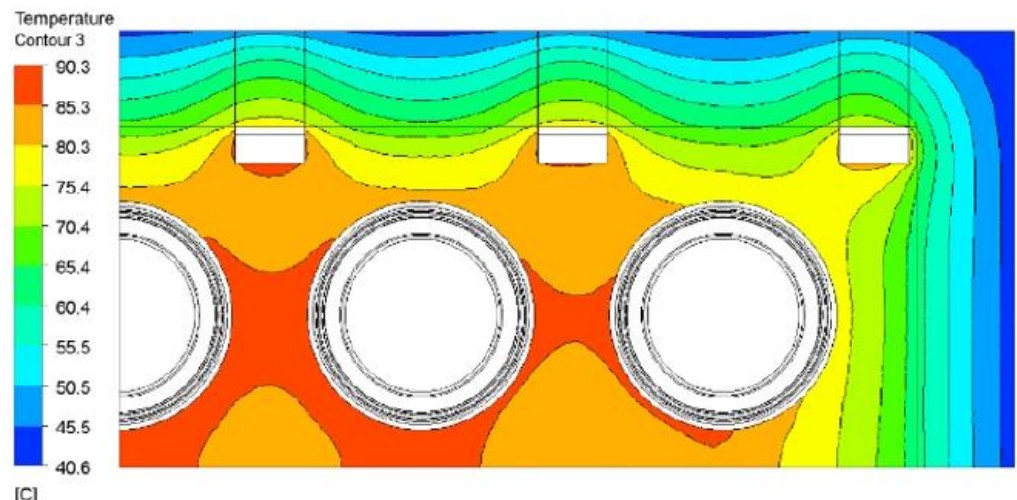

**Figure 9.** Temperature results under normal conditions at the upper wall.

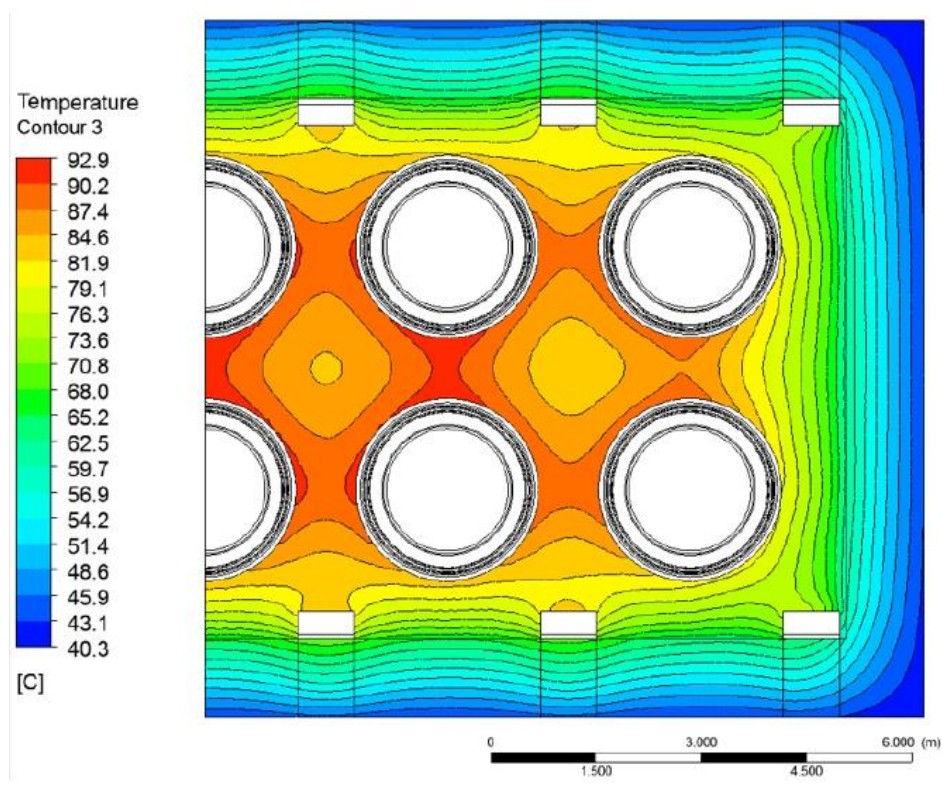

**Figure 10.** Temperature results under off-normal conditions at the upper wall.

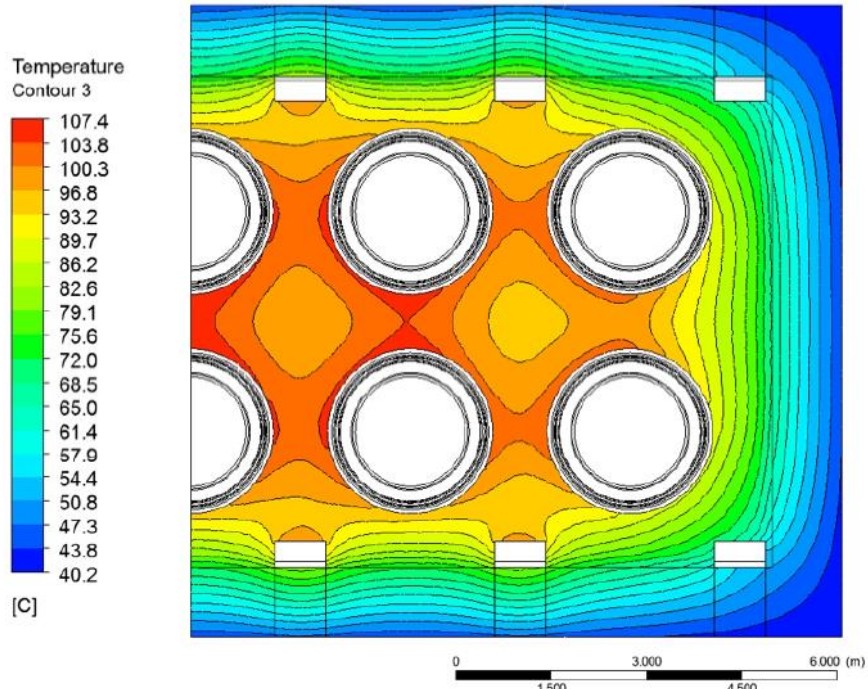

**Figure 11.** Temperature results under accident conditions at the upper wall.

### 3.5. Structural Analysis

The DSM is a reinforced concrete structure with a height of 7155 mm, a width of 9920 mm, and a length of 20,500 mm, and the module structure is installed on the sub-base of the bottom. DSM structural analysis includes static, thermal stress, and seismic analysis, and load conditions for each should be established and used for said analysis. After analyzing each load condition using ANSYS 19.2 [18]—a commercial analysis program —the loads were combined as shown in Table 4, according to ACI 349-01 [16] and NUREG-1536 [9]. The dead load was used for calculation after summing the masses of various structures, and the live load was considered to be the snow load and the load generated on the structure during SNF handling. The wind load was considered to be 60 m/s of wind speed, according to KEPIC [19], and the temperature distribution of the thermal load was derived as a result of thermal evaluation. Seismic load was applied with a maximum acceleration corresponding to 0.3 g, which is a safe shutdown earthquake (SSE) standard. The maximum acceleration was derived using the acceleration time history conforming to the design response spectrum of US NRC RG 1.60 [20]. Accident load was applied as the load generated by the impact of an SNF transport vehicle with a load of 200 tones at a speed of 8 km/h. Figure 12 shows the structural analysis results for load combination 8 with maximum stress in off-normal conditions, while Figure 13 shows the structural analysis results for load combination 16 with maximum stress in accident conditions. The structural analysis results obtained by combining each load are summarized in Table 4. As a result of structural analysis, it was confirmed that the structural integrity was maintained, as the maximum stress was determined to be below the allowable stress for all load combinations.

**Table 4.** Load combinations for structural evaluation of DSM.

| Condition | Load Combination | Ref. | Maximum Stress (MPa) | Allowable Stress (MPa) |
|---|---|---|---|---|
| Normal | (1) 1.4 D [1] + 1.7 L [2]<br>(2) 1.4 D + 1.7 L + 1.7 W [3]<br>(3) 0.9 D + 1.7 W | ACI-349 | 6.27<br>6.32<br>4.01 | |
| Off-normal | (4) 1.05 D + 1.3 L + 1.05 To [4]<br>(5) 1.05 D + 1.3 L + 1.3 W + 1.05 To<br>(6) 1.05 D + 1.275 L + 1.275 To<br>(7) 1.05 D + 1.275 L + 1.275 W + 1.275 To<br>(8) 0.9 D + 1.275 W + 1.275 To | ACI-349<br>NUREG-2215 | 17.51<br>17.51<br>21.81<br>21.82<br>22.4 | 30 |
| Accident | (9–11) 1.0 D + 1.0 L + 1.0 To + 1.0 Ess [6]<br>(12–14) 0.9 D + 1.0 To + 1.0 Ess<br>(15) 1.0 D + 1.0 L + 1.0 Ta [5]<br>(16–18) 1.0 D + 1.0 L + 1.0 Ta + 1.0 Ess<br>(19–21) 0.9 D + 1.0 Ta + 1.0 Ess | ACI-349 | 17.17<br>17.26<br>26.2<br>27.02<br>27.01 | |

[1] D: dead load, [2] L: live load, [3] W: wind load, [4] To: normal thermal load, [5] Ta: transient thermal load, [6] Ess: seismic load.

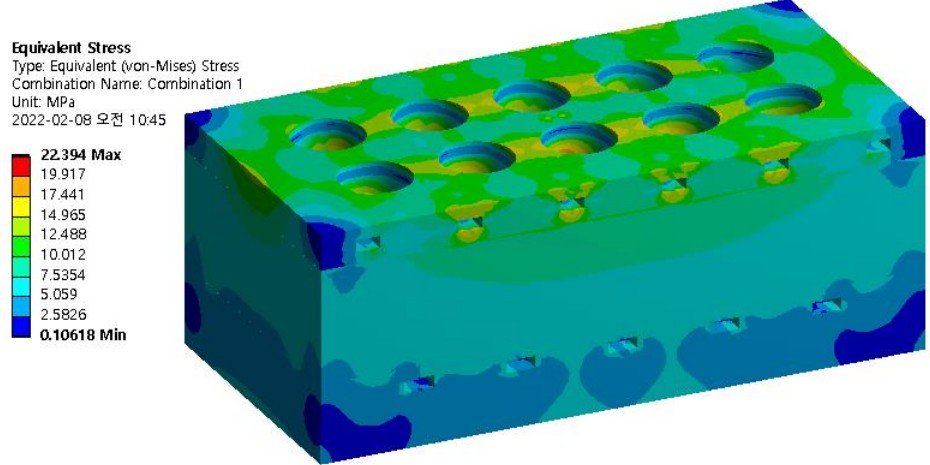

**Figure 12.** Maximum stress of load combination 8.

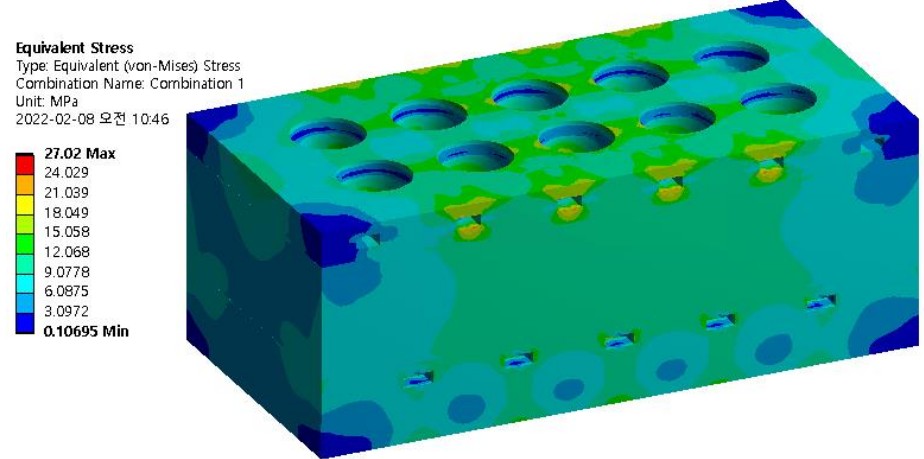

**Figure 13.** Maximum stress of load combination 16.

## 4. Conclusions

A newly developed modular dry storage system for PWR SNF was proposed, and air inlets and outlets were arranged at the top and bottom of the structure to enable non-powered natural convection cooling in a square-box-type concrete structure. The

DSM cylinder was installed between the canister and the external air layer, so that the stainless steel material of the canister does not contact external air. This structure can secure soundness for long-term storage, as there is no corrosion effect due to chlorine ions from the outside air. In addition, the storage method using a canister is advantageous in linking to the next management step, because when delivering SNF from a storage facility in an NPP to an intermediate storage facility, canisters containing SNF can be directly transported without repackaging.

Based on these advantages, the conceptual design of the DSM was developed for future application. To investigate the applicability of the conceptual design, source terms, criticality, radiation shielding, thermal management, and structural evaluation were performed in accordance with the design requirements. The reference fuel for evaluating the integrity of the dry storage module was PLUS7, and a fuel with a maximum initial concentration of 5.0 wt.%, a maximum average burnup of 45GWD/MTU, and a minimum cooling period of 10 years was selected.

As a result of performing the safety evaluation for each field of the reference fuel, it was confirmed that the developed DSM meets the design requirements in terms of the criticality aspect, and that the shielding performance of the module structure was sufficient. The thermal evaluation was performed below the temperature limit to ensure the integrity of the concrete and fuel, but it was thought that additional margins would need to be secured for the effective operation of the facility. In particular, the biggest load generation factor in the structural analysis was also thermal load. Therefore, in order to secure the temperature margin of the DSM, we concluded that the convective heat transfer performance can be improved by controlling the specifications and material of the insulation plate, the distance between the cylinders, the helium filling between the cylinder and the canister, and the shape of the outer pin of the cylinder.

These evaluation results will be used for the detailed design of the dry storage module for future applications, including the preparation of a safety analysis report and experimental verification, such as thermal testing of a scaled-down model for licensing applications.

**Author Contributions:** Writing—original draft preparation, T.K.; formal analysis, K.K., D.L. and T.N.; conceptualization, S.C.; writing—review and editing, Y.K. All authors have read and agreed to the published version of the manuscript.

**Funding:** This research received no external funding.

**Institutional Review Board Statement:** Not applicable.

**Conflicts of Interest:** The authors declare no conflict of interest.

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
