# Peer review of "Conceptual Design, Development, and Preliminary Safety Evaluation of a PWR Dry Storage Module for Spent Nuclear Fuel"

_applsci, doi:10.3390/app12094587_

Round 1

Reviewer 1 Report

See attached file.

Author Response

Q1. In the last sentence of the abstract you say “This evaluation results will be used as basic data for detailed design of the dry storage module in the future to proceed with further research.”. Which kind of future research? I think you can ben more detailed on the development of your work in the next stages.
Response to Q1: I revised the contents as follows in abstract and conclusion.
This evaluation results will be used as basic data for detailed design of the dry storage to proceed with further research including the preparation of a safety analysis report and experimental verification like thermal test of scale-down model for licensing applications.

Q2. In the introduction, the contextualization of the topic is not well defined and supported by references. I suggest to be more clear in the description of the context and add some references.
Response to Q2: Edited the introduction as advised.

Q3. You cite “requirements of related regulations of Korea Nuclear Safety Act”, could you add some reference on this or detailed the main requirements? 
Response to Q3: This reference was added in the revised manuscript.

Q4. In table 1, could you add the specific reference for each considered requirement?
Response to Q4: The specific reference was added in the revised manuscript.

Q5. In the paragraph 2.1, you state that “the main requirements for dry storage of SNF are in accordance with the relevant regulations of Korea Nuclear Safety Act, IAEA SSS No.SSG-15 [3] and US 10CFR72. Could you explained better the choice that you made in order to identify the requirements? Have you selected the most conservative one or all the references report the same requirements? Please, be more clear and explicit to define the requirements.
Response to Q5: I revised the contents as follows.
The dry storage system for SNF must be designed to maintain the safe storage conditions of SNF, prevent damage of SNF, assure SNF retrievability and withstand the impacts of design-basis conditions such as normal, off-normal, and accident conditions.

Q6. In the section 2.2, you are qualitative in the description, however you present you DSM without comparison with other solutions. Could you add information also from a quantitative viewpoint in order to support your statements and your work?
Response to Q6: I revised the contents as follows.
The horizontal concrete module or vertical concrete cask currently being commercialized occupy a larger storage area than the vertical concrete storage module. On the other hand, the vertical concrete storage module has about 30% advantage of reducing the storage area due to these structural characteristics. It is advantageous for maintenance by applying a replaceable design to the cylinders installed in the module.

Q7. Please add a reference to ORIGEN-S module of SCALE 6.1.
Response to Q7: This reference was added in the revised manuscript.

Q8. In the section 3.2, you talk about validation of the model. Could you add some additional comments and results on this point?
Response to Q8: Added some additional sentences and equation as advised.

Q9. In table 2, you call the second column and the last row as Keff. Is this correct?
Response to Q9: I revised the content as follows.
SCALE codes, bias and uncertainty were assessed as 0.00749 and 0.01079 respectively

Q10. Please add a reference for MCNP code.
Response to Q10: This reference was added in the revised manuscript.

Q11. In the legend of Figure 8, you put into parenthesis 1, 2, 3. I suppose these numbers corresponds to the locations, could you be clearer? In addition, I think that the comments to the figure 8 could be integrated.
Response to Q11: The locations of parenthesis 1,2,3 in Figure 8 have added to Figure 1

Q12. Please add a reference for Fluent 19.2 code.
Response to Q12: This reference was added in the revised manuscript.

Q13. In the section 3.4, you have made a CFD analysis. Which convergence approach have you followed to be sure that your results have reached the convergence?
Response to Q13: I revised the content as follows.
In this CFD calculation, RMS residual levels for mass and momentum was considered to converge to 1E-4 and energy to 1E-06 or less, respectively.

Q14. Please add a reference for ANSYS 19.2 code?
Response to Q14: This reference was added in the revised manuscript.

Q15. In the section 3.5, add the reference to ACI 349-01 and NUREG-1536.
Response to Q15: This reference was added in the revised manuscript.

Q16. In table 4, add the reference as reported in the reference section, and not only the name of the reference.
Response to Q16: These references were added in the revised manuscript.

Q17. In the section 3.5, I think that the description of the developed model is not clear.
Response to Q17: I added description for the developed model as follows.
DSM is a reinforced concrete structure with a height of 7,155 mm, a width of 9,920 mm, and a length of 20,500 mm, and the module structure is installed on the sub-base of the bottom.

Q18. The reference section can be improved, as commented in previous points
Response to Q18: Edited the reference section as advised.

Reviewer 2 Report

Thanks for sharing your work. Please find my comments attached in a .pdf file.

Author Response

Q1. Line 18: “.[1]” should be “[1]
Response to Q1: This typo was corrected in the revised manuscript.

Q2. Line 8-9: “As a result,…” first part of the sentence is redundant (stated in last sentence). This can be improvised to: “As a result of the evaluation, all the requirements…”
Response to Q2: Edited the sentence as advised.

Q3. Lines 20-23: Too long to read. Please divided this sentence into multiple sentences.
Response to Q3: Edited the sentence as advised.

Q4. Lines 31-34: The motivation is not coming out as it should and can be stated with more confidence. For example, restating this to: “In order to design PWR SDSS, this work presents preliminary evaluation carried out in accordance with … US 10CFR72”
Response to Q4: I revised the content as follows
The goal of this study is to evaluate the feasibility of conducting detailed design through preliminary evaluation of critical, shielding, thermal and structural fields for the conceptual design of DSM. In order to design PWR SDSS, preliminary evaluation was carried out in accordance with the requirements of related regulations of Korea Nuclear Safety Act [5] and US 10CFR72[26].

Q5. Line 32: Please remove “of critical, radiation shielding a, thermal and structural area”—this is redundant as previous statement already summarized the requirements.
Response to Q5: Unnecessary parts were removed from the sentence.

Q6. Introduction section: A short literature review is needed. Can some previous works in safety evaluation of storage module be provided? If any, this can also quickly summarize various storage methods and/or pertinent references?
Response to Q6: I added literature review with patent for DSM in introduction section. 

Q7. Line 24: Please provide any pertinent reference to M/K-400 module? Are there any relevant publications/website links?
Response to Q7: This reference was added in the revised manuscript.

Q8. Table 1: Define FA and SSE prior to its usage. I see SSE is defined on page 10, but this should happen here.
Response to Q8: Edited the sentence as advised.

Q9. Line 66: Rephrase “a lot of.” For example, “storage module requires relatively significant handling area…”
Response to Q9: This typo was corrected in the revised manuscript.

Q10. Line 67: “On the other hand, the vertical dry storage module the…” is not grammatically correct?
Response to 10: This typo was corrected in the revised manuscript.

Q11. Line 71: Is there any particular reason why not other two/more sides also provide air
flow/inlet/outlet? Will that improve the cooling efficiency?
Response to Q11: I revised the content as follows
The upper part of the DSM has no air inlet/outlet to secure operational convenience and the short-side to minimize interference due to multiple arrangement.

Q12. Figure 1: The blue arrow should be inlet (instead of outlet)?
Response to Q12: This typo was corrected in the revised manuscript.

Q13. Line 84: “…structural, thermal,….analysis” is redundant (stated in last sentence). Simply say:“includes these evaluation results.”
Response to Q13: Unnecessary parts were removed from the sentence.

Q14. Line 97, 161,192, 210: Please provide references to all the software/codes (MCNP, SCALE, etc.)
Response to Q14: This reference was added in the revised manuscript.

Q15. Figure 3: Is this activity for one fuel assembly? Please clarify.
Response to Q15: Edited the sentence as advised.

Q16. Figures 2 and 3: These can be combined with one figure having Activity on secondary axis? This will save space, improving article’s professional look.
Response to Q16: These two figures are combined in the revised manuscript.

Q17. Line 140, the uncertainty matches the value in Table 2, not the bias?
Response to Q17: I revised the content as follows.
SCALE codes, bias and uncertainty were assessed as 0.00749 and 0.01079 respectively

Q18. Line 164: What were the flux-to-dose conversion factors used in MCNP model? Please provide details and pertinent references. This is integral for a reader choosing to replicate your work.
Response to Q18: This reference was added in the revised manuscript.

Q19. Table 3: Just to clarify the air flow was not modelled in MCNP? Or was it?
Response to Q19: MCNP cannot be modeled for air flow. However, the measurement location was modeled as an air environment.

Q20. Figures 5-7: Many dimensions are barely visible on the print? Can a high-resolution picture be embedded in the manuscript? This is especially true for Figures 76-7.
Response to Q20: I replaced the high-resolution Figure

Q21. Figure 8 caption: this should be total dose rate?
Response to Q21: Since the effect of neutron dose rate is insignificant, it is a value for the gamma dose rate

Q22. Line 186: temperatures 400/570 already mentioned in paragraph above. Please avoid redundancy.
Response to Q22: Unnecessary parts were removed from the sentence.

Q23. Line 194: What was the insulation material? Please provide.
Response to Q23: Insulation material is the foam glass. This material’s name is added in the revised manuscript.
.
Q24. Figures 10-12: Please provide temperature units. This can be mentioned in each caption.
Response to Q24: These captions were added in the revised manuscript.

Q25. Figures 13-14: color scale and its text is blurry on print. Please enlarge and provide high resolution image.
Response to Q25: I provided high resolution images.

Q26. Line 256: Do the authors plan to validate the simulation results with an experiment. I am particularly curious as in many cases simulation may underestimate parameters. For example, neutron dose rates are highly likely to deviate. How can it be ensured the modeled concrete is exactly the same as the one used in the simulation? Any change in density in composition will change the evaluation result(s)?
Response to Q26: I revised the content as follows.
These evaluation results will be used for the detailed design of the dry storage module for future applications including the preparation of a safety analysis report and experimental verification like thermal test of scale-down model for licensing applications.

Reviewer 3 Report

Overall an excellent piece of work.  I have a few comments.

On line 115 you refer to the "American Atomic Energy Society".  I think this should probably be the American Nuclear Society.

It would be useful to report uncertainties on the dose rate through the inlet and outlet ports as MCNP does a very poor job of calculating this type of transport.

Somehow the 5% enriched fuel and 45 GWD/MTU do not seem to be characteristic of PWR fuel.  The fuel may start out 5% enriched and achieve 45 GWD/MTU, but a PWR fuel that has achieved 45 GWD/MTU is not likely to still be 5% enriched.  Maybe that is not what is meant.  Then a better description of the fuel state would be useful.  

In the conclusions the thermal load is identified as having the smallest temperature margin.  It would be better to discuss this issue in the body of the paper and then refer to it in the conclusion rather than first mentioning it in the conclusion.

Author Response

Q1. On line 115 you refer to the "American Atomic Energy Society". I think this should probably be the American Nuclear Society.
Response to Q1: This typo was corrected in the revised manuscript.

Q2. It would be useful to report uncertainties on the dose rate through the inlet and outlet ports as MCNP does a very poor job of calculating this type of transport.
Response to Q2: The uncertainty about the dose rate is added to Table 3

Q3. Somehow the 5% enriched fuel and 45 GWD/MTU do not seem to be characteristic of PWR fuel. The fuel may start out 5% enriched and achieve 45GWD/MTU, but a PWR fuel that has achieved 45 GWD/MTU is not likely to still be 5% enriched. Maybe that is not what is meant. Then a better description of the fuel state would be useful.
Response to Q3: I revised the content as follows
Table 1: Initial Enrichment

Q4. In the conclusions the thermal load is identified as having the smallest temperature margin. It would be better to discuss this issue in the body of the paper and then refer to it in the conclusion rather than first mentioning it in the conclusion.
Response to Q4: I revised the content as follows.
However, under normal conditions, the local temperature of 90.3°C in concrete is less of a safety margin than the design criteria. For effective operation of DSM, it is necessary to secure additional safety margin by improving heat transfer performance.

Reviewer 4 Report

Minor phasing of the English in 1st PG "In Korea as well, since there is uncertainty about of operation of 21 SNF management facilities and the year of losing full core reserve is coming soon," Needs rewording as doesn't make sense in English. 

Section 2: Minor inconsistencies between the off-normal/ normal conditions being assessed. For example the Exec summary mentions 'missiles' though terror or 3rd party attack is not covered in section 2.1. It should be noted that analysis of an 'attack' on the facility is not presented.

Section 2: The scope of the paper is not entirely clear. The authors are evaluating the DSM conceptual design basis to determine if it meets the requirements set out in Table 1. This is not explicit. Section 3 is equally confusing - a schematic is presented and it is not clearly presented what the actual purpose of the research is. It should clearly state that Figure 1 is the design concept and the authors are performing a series of calculations to assess it. These should be very explicit defined up-front. This is a criticism of the illogical way the work is presented rather than a gap in the work itself.

Section 3.1 minor English "The Korea" doesn't need 'the' in English.

Section 3.3. Too few details of the calculation are presented. For example, how was the credibility of the models confirmed?

Section 3.4: Again too few details presented. For example, was a mesh sensitivity analysis undertaken? What parameters were chosen for the heat transfer properties of the materials. What heat transfer mechanisms were modelled convection yes, radiation? If not say why.

Author Response

Q1. Minor phasing of the English in 1st PG "In Korea as well, since there is uncertainty about of operation of 21 SNF management facilities and the year of losing full core reserve is coming soon," Needs rewording as doesn't make sense in English.
Response to Q1: I revised the content as follows.
Similarly with other countries, there are uncertainty about the operation of SNF management facilities. Also as the year of losing full core reserve is coming soon, dry storage of SNF in NPPs sites should be promoted in a timely manner in preparation with the following factors: consideration of the stable operation of decommission of NPPs and the optimal SDSS.

Q2. Section 2: Minor inconsistencies between the off-normal/ normal conditions being assessed. For example the Exec summary mentions 'missiles' though terror or 3rd party attack is not covered in section 2.1. It should be noted that analysis of an 'attack' on the facility is not presented.
Response to Q2: This word of ‘missiles’ was removed from the sentence. Since this paper is about the conceptual design of the DSM, and the soundness of the missile attack will be evaluated later, the phrase has been excluded.  

Q3. Section 2: The scope of the paper is not entirely clear. The authors are evaluating the DSM conceptual design basis to determine if it meets the requirements set out in Table 1. This is not explicit. Section 3 is equally confusing – a schematic is presented and it is not clearly presented what the actual purpose of the research is. It should clearly state that Figure 1 is the design concept and the authors are performing a series of calculations to assess it. These should be very explicit defined up-front. This is a criticism of the illogical way the work is presented rather than a gap in the work itself.
Response to Q3: This paper was written focusing on the conceptual design of DSM and the preliminary evaluation required for it. Therefore, not all safety evaluations to secure the integrity of the SNF, but only the validity of the current concept design was performed. In the subsequent detailed design study, various evaluations including the analysis of accidents such as earthquakes, fires, and impacts will be additionally performed. The purpose of this paper is additionally described in the outline of section 3.

Q4. Section 3.1 minor English "The Korea" doesn't need 'the' in English.
Response to Q4: This typo was corrected in the revised manuscript.

Q5. Section 3.3. Too few details of the calculation are presented. For example, how was the credibility of the models confirmed?
Response to Q5: The uncertainty about the dose rate is added to Table 3

Q6. Section 3.4: Again too few details presented. For example, was a mesh sensitivity analysis undertaken? What parameters were chosen for the heat transfer properties of the materials. What heat transfer mechanisms were modelled convection yes, radiation? If not say why.
Response to Q6: I revised the content as follows.
The heat transfer inside the DSM was evaluated by CFD tools considering all the conduction, convection, and radiation heat transfer mechanisms. To study the effect of mesh resolution and near-wall treatment, three different computational meshes were constructed and evaluated, and a medium-size mesh was selected.

Round 2

Reviewer 1 Report

Authors try to improve the work, I appreciate their effort. However, I think that the paper is not suitable to be publish in the present form.

It continues to be not clear which is the aim of the work, and a clear descriprion of the novelty of your work is missing in the introduction.

The model should be better describe, the convergence of the model as been made applying which kind of method? You report only the convergence limit, but not the following approach.

Author Response

Q1. It continues to be not clear which is the aim of the work, and a clear descriprion of the novelty of your work is missing in the introduction.

A1. I revised the contents as follows in introduction

Therefore, in this study, the basic design of SDSS was performed to secure the following safety functions during the storage period.

  • Prevention of radioactive material leakage 
  • Decay heat removal from SF using natural airflow
  • Maintaining subcriticality of SF
  • Minimization of unnecessary radiation exposure of workers and surrounding residents.
  • Minimize the operating area of dry storage facilities.

In the conceptual design of the dry storage module (DSM), the concept of concentrating and storing the SF inside the rectangular concrete structure was used to secure the safety function. DSM uses a canister-cylinder dual structure to prevent leakage of radioactive materials, and it is possible to secure passive heat removal performance, and minimize radiation exposure and operating area by sharing the flow area inside the concrete structure.

Q2. The model should be better describe, the convergence of the model as been made applying which kind of method? You report only the convergence limit, but not the following approach

A2. I revised the contents as follows in Section 3.4

The interpolation scheme for pressure and velocity used a SIMPLE algorithms and upwind scheme.

Reviewer 4 Report

The paper still lacks a clear description of why the work is being published. In the introduction we have a single sentence that states there is uncertainty in the management of SNF facilities (ok like what?). There is a statement that the design of the dry storage module has been patented (ok, so this is the novel aspect of the work?). I would have expected the introduction to have been substantially re-written to address the concerns of the referees. Specifically the authors need to clearly state what the work is about, what is novel so its value can be clearly determined. 

There are still some minor changes required with the English, particularly with the added text. For example the phasing of this sentence is confusing " However, under normal conditions, the local temperature of 90.3°C in concrete is less of a safety margin than the design criteria." Clearly 90.3 is less than the safety limit of 93 degrees, but what I think the authors are saying is that this is too close to the safety limit and further optimisation is required. I don't intend to point these all out. 

Author Response

Q1. The paper still lacks a clear description of why the work is being published. In the introduction we have a single sentence that states there is uncertainty in the management of SNF facilities (ok like what?). There is a statement that the design of the dry storage module has been patented (ok, so this is the novel aspect of the work?). I would have expected the introduction to have been substantially re-written to address the concerns of the referees. Specifically the authors need to clearly state what the work is about, what is novel so its value can be clearly determined.

A1. I revised the contents as follows in introduction

Therefore, in this study, the basic design of SDSS was performed to secure the following safety functions during the storage period.

  • Prevention of radioactive material leakage 
  • Decay heat removal from SF using natural airflow
  • Maintaining subcriticality of SF
  • Minimization of unnecessary radiation exposure of workers and surrounding residents.
  • Minimize the operating area of dry storage facilities.

In the conceptual design of the dry storage module (DSM), the concept of concentrating and storing the SF inside the rectangular concrete structure was used to secure the safety function. DSM uses a canister-cylinder dual structure to prevent leakage of radioactive materials, and it is possible to secure passive heat removal performance, and minimize radiation exposure and operating area by sharing the flow area inside the concrete structure.

Q2. There are still some minor changes required with the English, particularly with the added text. For example the phasing of this sentence is confusing " However, under normal conditions, the local temperature of 90.3°C in concrete is less of a safety margin than the design criteria." Clearly 90.3 is less than the safety limit of 93 degrees, but what I think the authors are saying is that this is too close to the safety limit and further optimisation is required. I don't intend to point these all out. 

A2. I revised the contents as follows in Section 3.4

However, since the local temperature of 90.3°C in concrete has a very small margin compared to the design standard under normal condition, it is necessary to improve the structure with more efficient heat removal performance through detailed design.

This manuscript is a resubmission of an earlier submission. The following is a list of the peer review reports and author responses from that submission.